# Is Structured Exercise Performed with Supplemental Oxygen a Promising Method of Personalized Medicine in the Therapy of Chronic Diseases?

**DOI:** 10.3390/jpm10030135

**Published:** 2020-09-19

**Authors:** Nils Freitag, Kenji Doma, Daniel Neunhaeuserer, Sulin Cheng, Wilhelm Bloch, Moritz Schumann

**Affiliations:** 1Department of Molecular and Cellular Sport Medicine, Institute of Cardiovascular Research and Sport Medicine, German Sport University, Am Sportpark Müngersdorf 6, 50933 Cologne, Germany; n.freitag@dshs-koeln.de (N.F.); w.bloch@dshs-koeln.de (W.B.); 2Sport and Exercise Science, College of Healthcare Sciences, James Cook University, Townsville, QLD 4811, Australia; kenji.doma@jcu.edu.au; 3Sport and Exercise Medicine Division, Department of Medicine, University of Padova, 35128 Padova, Italy; neunhaeuserer@gmail.com; 4Department of Physical Education, Exercise, Health and Technology Centre, Shanghai Jiao Tong University, Shanghai 200240, China; sulin.cheng@jyu.fi; 5Faculty of Sport and Health Sciences, University of Jyväskylä, 40014 Jyväskylä, Finland; 6The Exercise Translational Medicine Centre, Shanghai Center for Systems Biomedicine, Shanghai Jiao Tong University, Shanghai 200240, China

**Keywords:** exercise medicine, clinical exercise science, individualized exercise prescription FiO_2_, hyperoxia, oxygen therapy

## Abstract

**Aim**: This systematic review aimed to explore the literature to identify in which types of chronic diseases exercise with supplemental oxygen has previously been utilized and whether this type of personalized therapy leads to superior effects in physical fitness and well-being. **Methods**: Databases (PubMed/MEDLINE, CINHAL, EMBASE, Web of knowledge and Cochrane Library) were searched in accordance with PRISMA. Eligibility criteria included adult patients diagnosed with any type of chronic diseases engaging in supervised exercise training with supplemental oxygen compared to normoxia. A random-effects model was used to pool effect sizes by standardized mean differences (SMD). **Results**: Out of the identified 4038 studies, 12 articles were eligible. Eleven studies were conducted in chronic obstructive pulmonary disease (COPD), while one study included coronary artery disease (CAD) patients. No statistical differences were observed for markers of physical fitness and patient-reported outcomes on well-being between the two training conditions (SMD −0.10; 95% CI −0.27, 0.08; p = 0.26). **Conclusions**: We found that chronic exercise with supplemental oxygen has mainly been utilized for COPD patients. Moreover, no superior long-term adaptations on physical fitness, functional capacity or patient-reported well-being were found, questioning the role of this method as a personalized medicine approach. Prospero registration: CRD42018104649.

## 1. Introduction

Therapeutic supplemental oxygen is often administered in patients diagnosed with pulmonary or cardiovascular diseases as well as in emergency or intensive care settings, in order to counteract resting systemic and/or local oxygen desaturation [1,2,3]. Although physical exercise is recommended as a supportive therapy for the majority of chronic diseases [4], exercise-induced hypoxemia might occur faster in patients with reduced oxygen transport capacity or resting oxygen saturation [5], leading to a limited physical capacity. Patients diagnosed with pulmonary disease are particularly prone to exercise-induced hypoxemia due to an impaired ventilatory capacity and/or ventilatory-perfusion mismatch [5]. In addition, cardiovascular impairments [6,7], treatment-induced side effects like chemotoxicity [8] or disease-induced inflammation [7] might potentiate the risk of exercise-induced hypoxemia in patients diagnosed with other types of chronic diseases. Exercise hypoxemia, in turn, may offset improvements in exercise tolerance, thereby attenuating the potential effects of physical training and further exacerbating deconditioning, eventually increasing morbidity and mortality [9,10].

Increasing the inspired oxygen fraction (FiO_2_) by providing supplemental oxygen might be an effective method to increase the compliance and adherence to prescribed exercise programs of patients suffering from chronic diseases and, thus, may potentially be used within a personalized exercise medicine approach. It was previously shown that supplemental oxygen administered during exercise maintained arterial oxygen desaturation, decreased perceived exertion and acutely increased performance capacity both in healthy [11,12,13,14] and diseased populations [14,15,16,17,18,19]. Despite the maintained arterial oxygen saturation [11], current evidence suggests that enhanced exercise performance is likely induced by a maintained muscle [11] and/or cerebral oxygenation [20,21]. This, in turn, would lead to increased oxygen diffusion, possibly improving neural drive [22]. In addition, supplemental oxygen may counteract exercise-induced arterial hypoxemia, which is thought to have detrimental effects on aerobic performance [23,24] and may, therefore, be of special interest for patients already suffering from decreased ventilatory capacity. Supplemental oxygen also appears to improve dynamic hyperinflation, breathing pattern and dyspnoea [25], all of which may further enhance exercise performance [26]. Furthermore, recent evidence suggests that supplemental oxygen may counterbalance cardiac arrhythmias [3], improve lymphocytic DNA repair [27] and brain function [19], consequently affecting patient-reported well-being as characterized by improvements in quality of life [28], perceived exertion [29,30] and fatigue [14,28].

However, despite potential beneficial effects of supplemental oxygen as a personalized therapy, possible adverse effects need to be considered. Studies have shown that greater FiO_2_ (i.e., >0.6) or overexposure (≥24 h) may induce lung intoxications, accompanied by excessive increases in reactive oxygen species (ROS) [14,31,32]. Large increases in ROS may, in turn, increase cellular and/or tissue damage and induce apoptosis [1,31], all of which may lead to severe adverse events in critically ill patients. Moreover, it was previously shown that supplemental oxygen may cause respiratory depression by inhibiting the respiratory drive, also potentially causing a hypercapnic acidosis and increasing the risk of pulmonary oedema and fibrogenesis [3,14]. Some studies also suggest that supplemental oxygen may actually reduce cardiac output and stroke volume by perturbing hemodynamic functioning and causing cerebral and coronary vasoconstriction, also compromising organ perfusion [3,14,33]. Thus, care has to be taken also when applying short-term supplemental oxygen during exercise [14,34].

The acute effects of supplemental oxygen have previously been summarized in a number of systematic reviews addressing both healthy [11,12,13,34,35,36] and diseased individuals, such as patients diagnosed with chronic obstructive pulmonary disease (COPD) [28,37,38], interstitial lung disease [17] or cystic fibrosis [15]. However, to the best of our knowledge, only two systematic reviews aimed at analysing the longitudinal effects of physical training with supplemental oxygen in patients diagnosed with chronic diseases. These reviews screened only studies that used oxygen therapy in COPD patients [39,40] and did not restrict the inclusion criteria to supplemental oxygen delivered solely during exercise [39]. Moreover, based on an initial generic search we identified papers that were not included in either of the two reviews, questioning the overall integrity of these findings.

Considering the theory that supplemental oxygen may counteract exercise-induced tissue hypoxia and improve perception and consequently exercise adherence, the primary aim of this systematic review was to explore the literature in order to assess whether chronic exercise with supplemental oxygen has been utilized also in diseases other than COPD. Our secondary aim was to assess safety, feasibility and efficacy of exercise performed with supplemental oxygen, in terms of physical fitness (i.e., cardiorespiratory fitness, peak power or muscle strength) and functional capacity (i.e., performance in shuttle walk test, six min walking test or stair climb test) as well as patient-reported well-being (i.e., quality of life or dyspnoea). By summarising the recent evidence of exercise performed with supplemental oxygen, we eventually aimed to extend our current understanding of personalized exercise medicine and potentially identify important gaps in research.

## 2. Methods

A systematic literature search was conducted in accordance with the Preferred Reporting Items for Systematic Reviews and Meta-Analyses (PRISMA) [41] and was registered with the international database of prospectively registered systematic reviews in health and social care (PROSPERO: CRD42018104649). The previous review by Pedersen and Saltin (2015) was used to characterize chronic diseases [4].

The electronic databases PubMed/MEDLINE, CINHAL, EMBASE, ISI Web of knowledge and Cochrane Library were systematically searched using the following search string, including medical subject headings, adapted to the specifications of each database: *((“Humans”[Mesh]) AND (((“Exercise”[Mesh]) OR “Exercise Tolerance”[Mesh]) OR “Exercise Therapy”[Mesh])) AND (((“Oxygen Inhalation Therapy”[Mesh]) OR “Hyperoxia”[Mesh]) OR “Oxygen/therapeutic use*”[Mesh])* (Appendix A). Relevant articles had to be published until 4th of December 2019. English and German language publications in human populations with no restriction to study design were included. Two authors (NF; KD) performed the literature search independently and, if needed, a third author (MS) provided further consultation. The search process entailed saving the online search, removing duplicates as well as consequently screening titles, abstracts and eligible full texts. Additionally, Google Scholar was searched for grey-literature and the reference lists of all potentially eligible full texts were screened.

### 2.1. Eligibility Criteria

The target population included male and female adults with at least one medical diagnoses of the following chronic conditions: cardiovascular, pulmonary, psychiatric, neurological, metabolic, musculo-skeletal or cancer (Figure 1). Supervised physical exercise training was defined as resistance- or endurance training as well as a combination of both with guidance from an exercise professional. Eligible studies were required to have reported the frequency, volume, intensity and type of exercise loading. Only training interventions with supplemental oxygen of at least three consecutive weeks were included, i.e., no acute bouts of exercise with supplemental oxygen were considered relevant for the purpose of this study. Supplemental oxygen was defined as inhaled gas with a higher oxygen content than 21% (normobaric) or increased oxygen partial pressure higher than one absolute atmosphere (hyperbaric). In fact, gas-mixtures that included a combination of different gases, such as helium and oxygen, were excluded in order to assess the specific effects of sole supplemental oxygen. Studies included were required to compare the effects of physical training with supplemental oxygen to that of a normoxia control group performing the same training intervention in ambient conditions (room air, medical air, compressed or humidified air), with no restriction to randomization (i.e., both RCT and CT were deemed eligible). Studies assessing sole hyperoxia/ambulatory oxygen therapy or hyperbaric oxygen therapy without an exercise component were excluded.

Primary endpoints of interest were the type of chronic disease in which exercise with supplemental oxygen was performed. Secondary outcomes of interest included reported safety (rated by clinically diagnosed adverse events and exercise- or supplemental oxygen-related dropouts) and feasibility (rated by adherence and compliance rates). Further secondary outcomes included markers of physical fitness, such as peak power (W_max_), peak oxygen consumption (VO_2peak_), time to exhaustion (TTE) and muscle strength (dynamic or isokinetic) as well as markers of functional capacity (i.e., distance in six-min walking test [6MWT] or time in shuttle walk test [SWT]) and patient-reported well-being, such as quality of life (QoL), dyspnoea and fatigue.

### 2.2. Data Extraction

The following data was extracted from each eligible full text: (a) general study information (first author’s last name, publication year, study design, study aim and outcome measures); (b) subject information (sample size, dropout rate, gender, age, type of chronic disease); (c) intervention data for administered supplemental oxygen and normoxia (i.e., general description, flow rate, device of delivery, oxygen concentration/partial pressure, compliance and effects) and (d) data on the exercise training program (general description, supervision, location, intensity, frequency, duration, start of intervention, follow-up period, compliance and effects). If data was missing or the data reporting was inconclusive, the authors of the included studies were contacted for clarification. Indications of any documented clinically diagnosed adverse events or exercise- as well as supplemental oxygen-induced dropouts were used to assess the safety of exercise training with supplemental oxygen. The feasibility was assessed by reported attendance or completion rates throughout the studies. Physical fitness was evaluated by identifying reported markers of exercise capacity, such as W_max_, VO_2peak_, time completed in TTE tests and responses in maximal heart rate (HR_max_) and blood lactate concentration (BLa). Furthermore, strength assessments via conventional dynamic or isokinetic measurements were considered relevant and included to assess physical fitness. Measures of functional capacity included distance covered in the 6MWT or SWT as well as stair climb tests. The recorded patient-reported well-being included questionnaire-assessed measures of QoL, anxiety, depression, dyspnoea and ratings of perceived exertion (RPE) at rest and during exercise as well as fatigue.

### 2.3. Data Synthesis and Analysis

In order to generalize the data, the number of parameters considered for pooled analysis had to be present in at least three studies reporting baseline and post-intervention values. Standardized mean differences (SMD) were calculated and a random-effects model was used to pool effect sizes using R (3.6.1) [42], RStudio (1.2.1335) [43] and the metafor package (version 2.2.1) [44]. The amount of heterogeneity (i.e., *τ*^2^), was estimated using the restricted maximum-likelihood estimator [45]. In addition to the estimate of *τ*^2^, the Q-test for heterogeneity [46] and the I^2^ statistic [47] were reported. In case any amount of heterogeneity was detected (i.e., *τ*^2^ > 0, regardless of the results of the Q-test), a credibility or prediction interval for the true outcomes was provided [48]. Studentized residuals and Cook’s distances were used to examine whether study results may be outliers and/or influential in the context of the model [49]. Studies with a studentized residual larger than the 100 × (1−0.05/(2 × *k*))^th^ percentile of a standard normal distribution were considered potential outliers (i.e., using a Bonferroni correction with two-sided α = 0.05 for *k* studies included in the meta-analysis). Studies with a Cook’s distance larger than the median plus six times the interquartile range of the Cook’s distances were considered to be influential. A trim-and-fill-contour funnel plot was provided to estimate the number of studies potentially missing from the meta-analysis [50,51,52]. The rank correlation test [53] and the regression test [52], using the standard error of the observed outcomes as predictor, were used to check for funnel plot asymmetry. To avoid a potential impact of baseline differences on the outcomes, the same procedure was applied for mean differences of the change from baseline to post-intervention measurements instead of solely post-exercise mean differences. Studies presenting median, range and the sample size were converted into estimated means and variance [54].

### 2.4. Assessment of Methodological Quality

The Cochrane Collaborations’ risk of bias assessment tool was used to evaluate the internal validity of the included randomized controlled trials (RCTs) [55]. Independently, two authors (NF; KD) examined the studies of interest for the following sources of bias: selection (sequence generation and allocation concealment), performance (blinding of patients/study-personal), detection (blinding outcome assessors), attrition (incomplete outcome data), reporting (selective reporting), and other potential bias (e.g., recall bias). Furthermore, the PEDro (Physiotherapy Evidence Database) scale was used to additionally assess the methodological quality of all included studies [56]. The PEDro scale contains eleven *yes* or *no* items. Criterion 2 to 9 are assessing the internal validity as well as the randomization. Criteria 10 and 11 evaluate if the study contains sufficient statistical information for interpretable and replicable results. Criteria 1 assesses the external validity and was added to ensure the completeness of the original Delphi list [57] but is not used to calculate the total PEDro score. Two reviewers (NF; DN) independently rated the included studies with agreement on every single item. Disagreement in ratings were discussed and if necessary, a third author (MS) was involved for decision. The PEDro score had to be ≥6 on the scale from 0 to 10 to be rated as a high-quality trial and a score of 4 to 5 to be considered of fair quality. Studies rated with ≤3 were considered of poor quality. Both rating tools were used to obtain a more comprehensive view of the included trials [58].

## 3. Results

A total of 4124 trials were identified. Removing duplicates and ineligible records led to 33 full texts for further assessment. Overall, 21 studies were excluded for methodological reasons (Figure 2), while one potentially eligible study [59] was excluded because it used the same study sample as another included study [60] and two other potentially eligible studies were excluded because the exercise intervention was not described in detail [61] or because non-invasive ventilation was used as control [62]. Finally, a total of 12 [60,63,64,65,66,67,68,69,70,71,72,73] articles were deemed eligible and analysed. Comprehensive information of individual study characteristics and study conclusions across the included studies are presented in Table 1 and in the Appendix A.

### 3.1. Study and Intervention Characteristics

All included studies were conducted as RCTs, with a total of 452 patients (mean age 64.6 ± 2.2 yrs.). The majority of studies included patients diagnosed with chronic obstructive pulmonary disease (COPD) [60,63,64,65,66,68,69,70,71,72,73] (94.9% of patients), while one study was conducted in patients with coronary artery disease (CAD) [67]. Therefore, no studies assessed the effects of exercise with supplemental oxygen in patients diagnosed with neither psychiatric, neurological, metabolic, musculo-skeletal nor oncological disease. An overall mean dropout rate of 21.9% (ranging from zero to 57.6%) led to a total of *n* = 353 analysed patients who were exercising with supplemental oxygen (*n* = 178; mean age 64.3 ± 3.2 yrs.; 32.0% women) and normoxia (*n* = 175; mean age 65.0 ± 3.6 yrs.; 30.4% women). Blinding was incorporated in eight studies using either a single blind approach [60,64,65] or double-blind design [66,69,71,72,73]. All studies were performed as parallel group comparisons except for one study, which used a crossover approach after six weeks of training [72]. However, to exclude potential carry over effects only the data assessed before the crossover was considered for the pooled analysis.

Supplemental oxygen was supplied with a mean flow rate of 4.6 ± 2.2 L per min, ranging from 2 to 10 L per min via compressed/gaseous oxygen. All included studies used normobaric supplemental oxygen. The reported FiO_2_ ranged from 0.35 [64,71] to 0.4 [69] and up to 0.6 [72], with some studies reporting an oxygen supply of 100%, without assessing the exact FiO_2_ [60,63,66,67,68]. In the majority of studies, oxygen was delivered through nasal cannulas connected either to an oxygen cylinder [63,65,66,71,72] or compressor [60,70,73], while four studies used a mask or mouthpiece connected to a Douglas bag [67,68,69] or a mixed chamber [64]. Normoxic control conditions included ambient air [63,67,68,70], ambient air delivered by a compressor [60,65,66,71,72] or humidified [69] decompressed ambient air [64] as well as medical air [73].

The duration of the exercise training interventions ranged from a minimum of four to a maximum of 24 weeks (mean duration 9.9 ± 5.7 weeks). The number of weekly training sessions ranged from one to five sessions per week (mean frequency 3.2 ± 0.9 sessions per week). The majority of studies utilized endurance training, either through a continuous loading [64,65,66,69,70,73] or intermittent exercise [60,63,67,68,72], with one study combining both types of training [71]. A total of nine studies solely incorporated endurance training [60,64,66,67,68,69,70,71,73], while two studies combined endurance training with functional tasks including resistance exercises and stair climbing [63,65]. Only one study conducted a concurrent training approach with endurance interval training and machine based resistance exercises [72]. The training intensity ranged from somewhat moderate (i.e., 80% 6MWT; 80% VO_2peak_; 75% W_max_) [65,71,72,73] to high (i.e., 80% W_max_; 95% HR_max_, 85% VO_2peak_; RPE 17) [60,64,66,67,68,69,70,71,72] and supramaximal [71] (i.e., 110–120% of W_max_), with one study not reporting the training intensity [63] (Table 1). The intensity of the resistance training ranged from lifting and carrying lightweight [63,65] to three minutes of isometric exercises [63] without further specification. One study used intensive machine-based resistance training with one set of 8 to 15 repetitions to failure [72]. The duration of the training sessions was 39.6 ± 9.7 min ranging from 30 [60,67,70,71] up to 60 [63,65] min with incomplete reporting of times for warm-up and cool-down (Table 1).

### 3.2. Risk of Bias Assessment and PEDro Scale Ratings

The methodological quality assessments of all included studies are summarized in Figure 3. An appropriate procedure for a randomly generated sequence was fully described in five studies [60,70,71,72,73] and information about allocation concealment was given in five studies [65,70,71,72,73]. Performance bias was unclear in four studies [63,67,68,70], while detection bias was found in three studies [60,64,65]. Attrition bias was present in two trials [67,70] and potentially influential in another five trials [63,64,71,72,73]. A possible reporting bias due to high dropout rates and incomplete data reporting was present in three studies [67,70,71], with unclear risk presented in seven studies [60,63,64,65,66,68,69]. The mean score on the PEDro scale across all studies was rated with seven. Therefore, the overall quality of the included studies was high, even though three studies were of fair quality [63,67,70] and one study of poor quality [68].

### 3.3. Feasibility and Safety

The results for safety and feasibility are limited to pulmonary and cardiovascular diseases, especially in COPD and CAD patients. Despite three studies not reporting dropouts [63,64,69], the remaining studies reported a total of 50 patient dropouts in the supplemental oxygen groups and 49 patient dropouts in the normoxia groups, respectively. These were related to exacerbations [65,70,72], hospitalizations [68], airway infections and illness [60,66,67], bone fracture [72], co-morbidities [71,72], withdrawal [71,72,73] and intervention-unrelated death [70,73], while in three studies no reason was specified [67,70,71]. In addition, there was no trend towards a higher rate of airway infections in either condition. Only two studies reported possible exercise-induced adverse events, i.e., exacerbations with supplemental oxygen and normoxia as well as atrial fibrillation [73], elevated ST exercise segment [67] with supplemental oxygen and mild stroke [73] in the normoxia group. Training adherence was only reported in three studies [66,67,70] and ranged from 89% to 100% with supplemental oxygen and 87% to 100% in normoxia.

### 3.4. Intervention Effects and Pooled Analysis

#### 3.4.1. Physical Fitness and Functional Capacity

An overview of the individual effects reported in the included studies comparing exercise with supplemental oxygen and exercise in normoxia is provided in the supplements (Appendix A). Physical fitness included the following parameters: W_max_ [63,64,66,67,68,69,71,72], VO_2peak_ [63,64,66,67,68,69,71,72], HR_peak_ [60,63,64,66,67,68,69,72], BLa [60,64,66,67,68,71,72], and TTE [63,66,69]. Reported parameters of functional capacity included distance covered in the 6MWT [60,63,71], the ISWT [65,73] and distance or time achieved in the ESWT [70,73], as well as the number of light weight (1–2 kg) lifts and stair climb tests [63] (Appendix A). Due to insufficient reporting of results or low number of studies assessing the required parameters, only W_max_, VO_2peak_ and 6MWT were included in the meta-analysis. The observed effects of post-intervention comparisons for W_max_ (SMD −0.30; 95% CI −0.60, 0.00; *p* = 0.05) pointed towards a statistical difference between conditions in favour of normoxia, while no statistical difference was observed for VO_2peak_ (SMD −0.11; 95% CI −0.40, 0.19; *p* = 0.49) and 6MWD (SMD −0.05; 95% CI −0.72, 0.62; *p* = 0.88) (Figure 4). When considering pre to post changes (Δ), no between-condition effects were observed for either of the variables assessed (ΔW_max_: SMD −0.03; 95% CI −0.33, 0.26; *p* = 0.83, ΔVO_2peak_: SMD 0.02; 95% CI −0.27, 0.32; *p* = 0.88 and Δ6MWD: SMD −0.05; 95% CI −0.49, 0.39; *p* = 0.82) (Figure 5). Based on individual studies, none of the physical fitness parameters showed a statistically significant difference between exercise with supplemental oxygen and exercise in normoxia except for one study, which found superior effects of increased FiO_2_ of 0.6 with a high flow rate of 10 L per min compared to the normoxia group in relative W_max_ [72] (Appendix A).

#### 3.4.2. Patient-Reported Well-Being

Inconsistent data reporting coupled with the fact that only a few studies have used similar tools to assess patient-reported well-being made it difficult to draw comparisons between the included studies. Patient-reported outcomes included RPE during rest and maximal exertion, ratings of dyspnoea and (leg) fatigue using either the Borg scale (6–20) [74] or the modified Borg scale (0–10) [75]. Furthermore, the following questionnaires were used: Chronic Respiratory Disease Questionnaire (CRDQ) [63,65,66,73]; Hospital Anxiety and Depression Scale (HADS) [65,72]; London Chest Activity of Daily Living Scale (LCADL) [65]; 36-Item Short Form Healthy Survey (SF-36) [66,67,71]; Macnew Heart Disease Health-related Quality of Life Questionnaire (Macnew) [67]; St. George’s Respiratory Questionnaire (SGRQ) [70]; Dyspnoea-12 questionnaire [73]. The meta-analysis of post-intervention comparisons for Dyspnoea (SMD 0.25; 95% CI −0.18, 0.67; *p* = 0.26) revealed no statistical difference between exercise with supplemental oxygen and exercise in normoxia (Figure 4). The pooled effects for ΔDyspnoea (SMD 0.07; 95% CI −0.33, 0.48; *p* = 0.72) confirmed these results and no statistical difference was observed between the two conditions (Figure 5). On an individual study level, between-group differences were detected for dyspnoea [65], and the Mastery subscale [66] (CRDQ) as well as the General Health scale in the SF-36 [66], with significant improvements for exercise in the supplemental oxygen group compared to the normoxia group (Appendix A).

#### 3.4.3. Publication Bias

The funnel plot did not show a clear funnel-shape across all assessed and pooled effects sizes (Figure 6). However, neither the rank correlation nor the regression test indicated any funnel plot asymmetry (*p* = 0.140 and *p* = 0.186, respectively). The visual observation provided by the trim-and-fill-function confirmed study heterogeneity, while potential publication bias and methodological heterogeneity remains possible, as indicated by a large cluster in the centre of the plot with no values in the top or bottom right and left corner, respectively.

## 4. Discussion

In times of an improved understanding of *exercise* as *medicine*, there is an increasing precision of therapeutic measures and the number of innovative therapy approaches is growing. Although in previous reviews the beneficial effects of exercise with supplemental oxygen in COPD patients has been questioned [39,40], supplemental oxygen has indeed been shown to induce beneficial effects in healthy populations [11] and, thus, it remains unknown whether this method may be a mean of personalized therapy for patients diagnosed with other types of chronic diseases. Our findings clearly expand on previous reviews [39,40] by including twelve studies that have utilized exercise with supplemental oxygen. However, while it was somewhat expected that the majority of studies identified were conducted in COPD patients [60,62,63,64,65,66,68,69,70,71,72,73], it was surprising that only one study included patients other than a chronic lung pathology [67]. Thus, despite numerous medical hypotheses and the previous findings obtained from healthy populations, we currently have no evidence for the application of training with supplemental oxygen in psychiatric, neurological, metabolic, musculoskeletal or oncological disease. Moreover, even though a large number of studies was included in this review, no statistical advantage for improvements in physical fitness, functional capacity and patient-reported well-being was found.

The pathogenesis of chronic diseases is generally characterized by proinflammatory processes, associated with a reduced exercise tolerance [76]. Typically, the disease-induced inflammation leads to hypoxic microenvironments, affecting not only oxygen transport and diffusion but also energy flux and immunoregulatory mechanisms [76,77]. Resulting chronic tissue hypoxia may further increase inflammation and deteriorate physical deconditioning and, thereby, contribute to the vicious cycle of chronic disease progression and exercise intolerance [77].Tissue hypoxia and inflammation, in turn, may be counteracted by regular physical exercise but training adherence is often compromised by adverse symptoms, such as dyspnoea and fatigue. This is why previous studies have aimed to assess the effects of exercise with supplemental oxygen in an attempt to reduce subjective exertion and breathlessness [29,78,79,80,81], while concomitantly enhancing tissue re-oxygenation [14,76,82]. Because patients suffering from pulmonary diseases may be more prone to ventilatory limitations both at rest and during exercise, it was not surprising that the majority of studies identified in this systematic review was performed in COPD. However, considering that fatigue and breathlessness are also severe symptoms in other chronic conditions, such as cancer or cardiovascular diseases, it was somewhat unexpected that only one study was conducted in patients suffering from a chronic heart condition [67], while no studies were performed with patients of other entities. Especially in light of the growing interest for more personalized therapeutic approaches for chronic diseases, we hypothesized that training with supplemental oxygen could be an effective way to overcome limitations and barriers to physical training and, thus, increase adherence and therapeutic success.

However, a main reason for a lack of studies applying supplemental oxygen during exercise training in other chronic diseases may be related to safety concerns. Previous studies have shown that chronic supplementation of oxygen administered for days or even weeks is associated with severe side effects, such as oxygen toxicity and pulmonary tissue damage [3,14]. Furthermore, cerebral, cardiovascular and pulmonary vasoconstriction as well as an increased cardiac resistance and lowered coronary blood flow were previously observed [14,15]. However, there are also studies indicating pulmonary vasodilatation and increased cardiac output, ultimately leading to an improved exercise capacity with supplemental oxygen [25,83,84]. Therefore, the heterogeneity of reported physiological findings might contribute to a rather conservative application of training with supplemental oxygen, as indicated by the overall low number of studies identified in our review. Unfortunately, hemodynamic parameters were assessed only in one study investigating the effects of supplemental oxygen during exercise on peak cardiac output and stroke volume [67]. In this study, it was shown that cardiac output and stroke volume increased to a similar extent both after supplemental oxygen and normoxic training [67], indicating that a short-term increase (i.e., only during exercise sessions) in FiO_2_ may not induce cardiovascular complications. However, because mechanistic markers were not assessed in either of the included studies, a thorough risk evaluation may not be performed at this stage.

Interestingly, the patient dropout rates were low to moderate throughout all included studies (i.e., 21.9% with supplemental oxygen and 21.8% in normoxia). Moreover, even in those studies with the highest dropout rates [70,71,72,73], no associations between the FiO_2_ (0.35 to 1.0) or flow rate (2 to 10 L min^−1^) and the number of dropouts were observed. In addition, the reported training adherence ranged from 89 to 100% for both conditions. However, even though the majority of studies was rated of high quality based on the PEDro scale, the overall reporting of adverse events as well as the rates for attendance and compliance of participants was not adequate across most of the included studies. Authors of future studies are directed to the common terminology criteria for adverse events (CTCAE) [85]. Thus, concluding remarks on the safety and feasibility have to be interpreted with caution.

Previous cross-sectional studies in COPD [37,86,87,88], cystic fibrosis [15,89] and interstitial lung disease [17] have well indicated that acute exercise with supplemental oxygen leads to transient increases in exercise capacity mainly through an increased oxygen partial pressure and saturation, improved breathing pattern and reduced dynamic hyperinflation [3,10,14]. Furthermore, studies assessing the effects of acute bouts of exercise with supplemental oxygen in patients with chronic heart failure [18,30] or diabetes mellitus type 2 [90,91] showed beneficial physiological effects such as increased exercise performance, reduced fatigue and breathlessness. Despite improved muscle oxygenation, acute supplemental oxygen may also cause an inhibition of carotid body stimulation as well as a reduced respiratory drive and pulmonary vasodilatation and, thus, may counterbalance exercise-induced hypoxemia and enhance exercise performance [25,83,84]. However, based on the results of our meta-analysis, the previously reported acute benefits of supplemental oxygen do not seem to translate into long-term adaptations. In fact, out of the included studies, only two studies reported improvements in markers of physical fitness, such as time to exhaustion or peak power [66,72], while two other studies showed greater reductions on exercise-induced dyspnoea [65,73]. As such, even when including a larger number of studies as compared to previous reviews with patients [39,40], presently there is no clear evidence for exercise being performed with supplemental oxygen to induce clinically meaningful adaptations in either biological or performance markers but these findings are limited to COPD and CAD patients.

A number of methodological aspects may explain the heterogeneous study findings. First, an important consideration for exercise training with supplemental oxygen is the potentially lowered perceived exertion, along with an increased exercise capacity [16,30,92]. Consequently, supplemental oxygen may allow to sustain a given power output for longer durations or to perform at higher training intensities. Therefore, it might be necessary to adjust the training intensity for this acute enhancement of exercise capacity to avoid an actual lower exercise load [93]. Interestingly, in the majority of included studies an adjustment for this phenomenon was not considered. In fact, only one study clearly controlled for the exercise intensity but this work was among the few studies that have actually shown superior effects on physical fitness [72]. It is, thus, likely that the lack of beneficial effects in the supplemental oxygen groups in most of the remaining studies was also due to a relatively lower training intensity for supplemental oxygen conditions, emphasizing the need for personalized exercise prescription.

In addition, also technical aspects of oxygen administration need to be considered. For example, supplemental oxygen may be provided humidified or dry, liquid or gaseous and supplied in normobaric or hyperbaric conditions [93,94,95,96,97]. Furthermore, oxygen concentrations typically range from 30 to 100%, with flow rates of 2 to 20 L per min [11,12,93]. It is obvious, that the variety of devices and methods used for oxygen delivery bears difficulties for comparison and generalizability of findings described in individual studies included in this review. In fact, it was shown in earlier cross-sectional studies including COPD patients that a high-flow rate between 3 to 8 L per min increased distance covered in walking tests around 2.4 times, compared to a much lower flow of only 0.5 to 4 L per min [98]. Furthermore, it was previously reported in patients with severe airflow obstruction that cycling time may be increased by 51%, 88% and 80% with concomitant flow rates of 2, 4 or 6 L per min, respectively [99], supporting the need to adjust exercise intensities/durations during exercise training with supplemental oxygen. A similar dose-response relationship also seems to exist for the FiO_2_ and time to exhaustion, which progressively increased with higher FiO_2_ up to 0.5 [11,25]. However, because of the heterogeneity in the methods used to deliver oxygen in the included studies, similar associations were not observed in the present review.

Nonetheless, both the flow-rate and FiO_2_ are dependent on whether compressors or gas cylinders are used and whether the oxygen is supplied through nasal cannulas, rebreather masks or oxygen on demand systems [3,100], further complicating comparisons of different studies. Patients might prefer the widely used nasal cannulas for oxygen delivery instead of facemasks due to a higher comfort and lower facial constriction. Interestingly, it was previously shown that reported FiO_2_ values using nasal cannulas to deliver oxygen can be misleading due to an actual high variability in delivery rate [101,102]. This might become even more relevant while patients are exercising, due to changing breathing and ventilatory patterns with increasing exercise loads. For example, patients typically switch from nose breathing to oronasal or mouth breathing with increasing exercise intensity [103], even during submaximal exercise [103]. Thus, it is questionable how much oxygen is actually inhaled by the patients using nasal cannulas. This is an important consideration because the majority of studies identified in our systematic review indeed used nasal cannulas for oxygen delivery [60,63,65,66,71,73,104], while only few of the remaining studies administered oxygen either through a mask [64,67,68] or mouthpiece [69] or did not specify the delivery system [70]. However, despite the limitations of nasal cannulas, we were not able to identify differences in terms of training efficacy based on the system used and, thus, the significance of these technical aspects requires further assessment. However, contrary to these apparent limitations, these considerations also bear the potential to further personalize the use of exercise training with supplemental oxygen based on the individual needs. This also includes other promising approaches to counteract local and/or systemic hypoxia, such as the injection of oxygen nanobubbles [105,106,107].

Finally, the observed variations in the outcomes might be partially explained by the heterogeneous study populations and disease stages within the individual trials (i.e., hypoxemic, normoxemic, GOLD 1–4 COPD). Interestingly, the only two studies showing beneficial effects of supplemental oxygen on physical fitness involved nonhypoxaemic COPD patients [66,72]. The remaining studies included either exercise-hypoxaemic [60,63,65], nonhypoxaemic [70,71,73] or CAD patients [67]. Considering the differences in symptom-severity, stage and pathophysiological impact of disease, aim of the exercise intervention, and rehabilitation setting, the question arises whether the one-size fits all approach in oxygen supplementation during training is reasonable. Future studies should, therefore, focus on a precise dose, timing and purpose of supplemental oxygen during exercise training rather than on the sole question of whether supplemental oxygen needs to be provided. Another important aspect might be that the reported exercise-related parameters in available studies might not necessarily be of clinically significance. Although strong correlations do exist between cardiorespiratory fitness and mortality [108], it cannot be ruled out that the chosen parameters in the included studies were insufficient to detect clinically meaningful changes induced by exercise training with supplemental oxygen, considering also rather short intervention periods of only 9.9 ± 5.7 weeks. Consequently, it is recommended to include assessments of underlying biological mechanisms into future investigations, as well as to further extent the duration of study protocols.

## 5. Conclusions

Exercise medicine as a part of personalized therapy provides a huge potential for diseased populations and, thus, innovative methods for various chronic conditions are warranted. The use of supplemental oxygen during exercise might at least theoretically bear the potential for an effective therapeutic approach to counteract chronic disease-induced inflammation and tissue hypoxia for literally all chronic diseases, while concomitantly increasing patient adherence and therapy compliance. However, as shown by this systematic review, so far the majority of studies assessing the effects of chronic exercise with supplemental oxygen were carried out in COPD patients, while we found only one study that was performed with CAD patients and currently no evidence exists for other chronic diseases. Our findings also indicate very scarce data concerning safety and feasibility of exercise with supplemental oxygen and this data is limited to COPD and CAD. Moreover, the interpretation of that data is somewhat hindered by incomprehensive reporting. Interestingly, even though it appears that supplemental oxygen is a common therapeutic method to support exercise interventions in COPD patients, our findings support previous reviews by clearly indicating that there is currently no evidence superior effects in terms of physical fitness, functional capacity or patient-reported well-being. While it is likely that this may be related to heterogeneous study designs and/or technical aspects of oxygen delivery (i.e., low flow rates and oxygen delivered through nasal cannula), there seems to be a gap between findings obtained from acute study designs and long-term interventions. Thus, future studies should aim at identifying dose-response relationships of supplemental oxygen delivered and further assess whether this type of training may be a beneficial part of a personalized medicine approach for other types of chronic diseases.

## Figures and Tables

**Figure 1 jpm-10-00135-f001:**
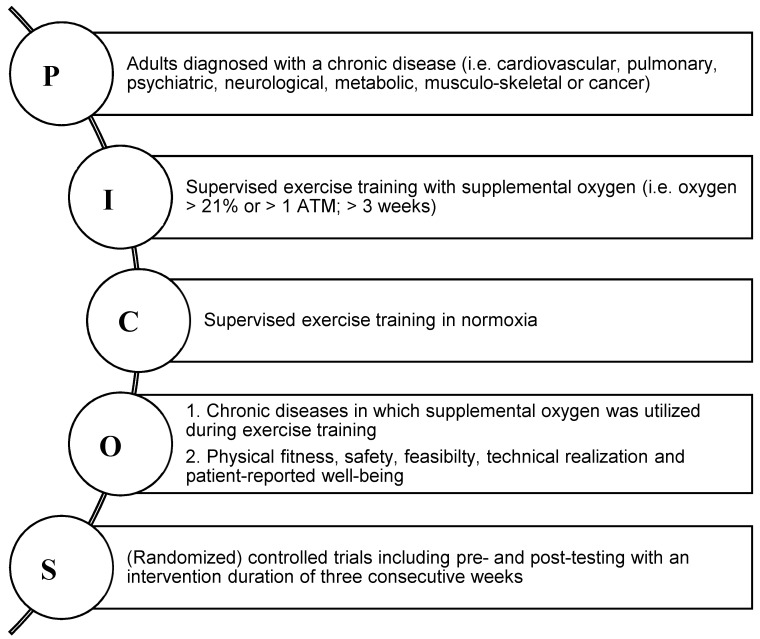
Inclusion criteria defined as PICOS; P-Population, I-Intervention, C-Comparison, O–Outcome, S-Study designs; ATM-Standard atmosphere pressure.

**Figure 2 jpm-10-00135-f002:**
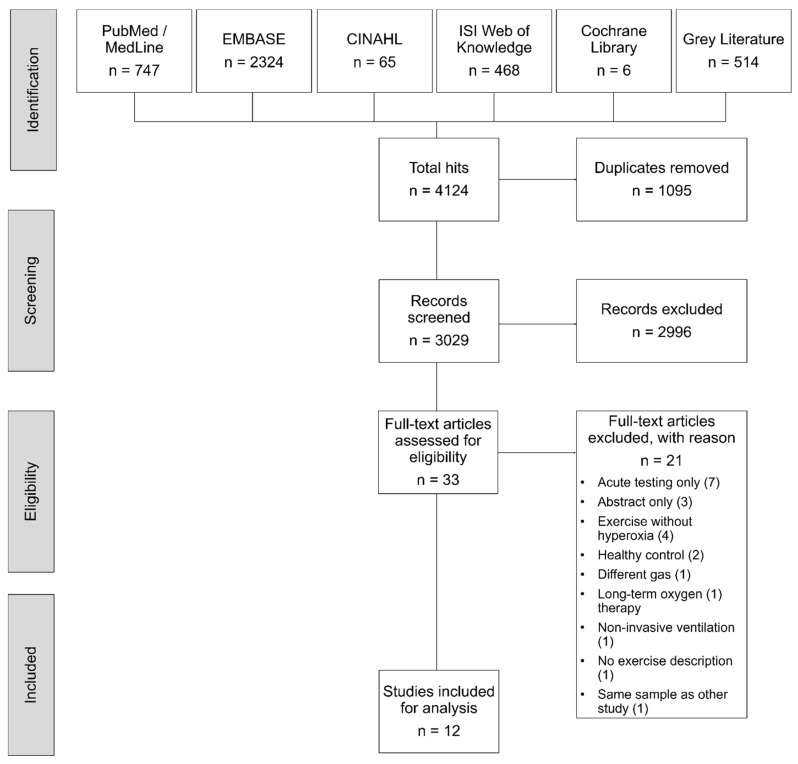
PRISMA flowchart of the systematic review process. PRISMA, Preferred Reporting Items for Systematic Reviews and Meta-Analysis.

**Figure 3 jpm-10-00135-f003:**
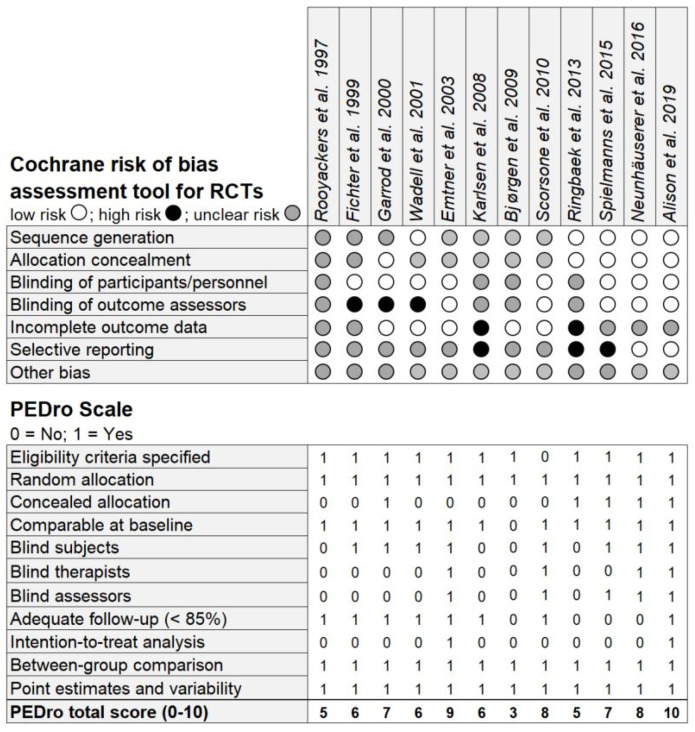
The Cochrane Collaboration’s tool for assessing risk of bias and PEDro scale from the Physiotherapy Evidence Database to determine quality of clinical trials. The total PEDro score is the sum of all criteria except *eligibility criteria specified*.

**Figure 4 jpm-10-00135-f004:**
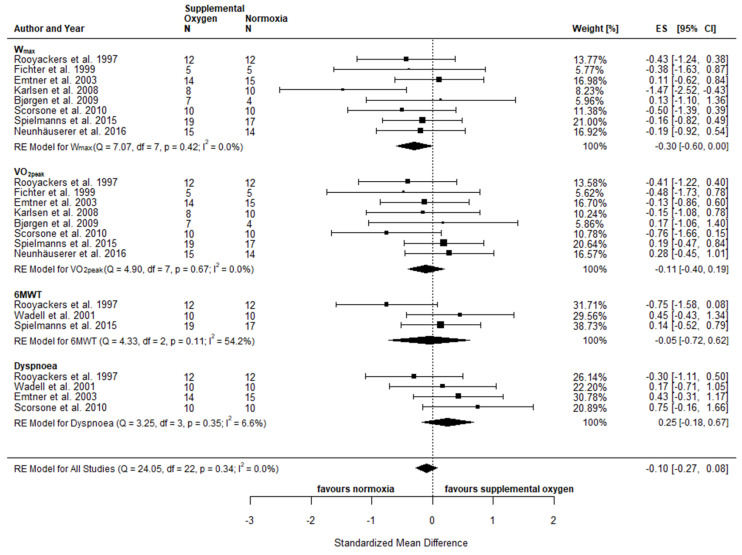
Physical fitness and patient-reported well-being comparing supplemental oxygen and normoxic exercise training interventions using absolute values of endpoint comparisons. CI confidence intervals; ES effects size Cohen’s d (corrected for small samples); df degrees of freedom; I^2^ and Q (Cochran’s Q) describe heterogeneity, RE random effects model.

**Figure 5 jpm-10-00135-f005:**
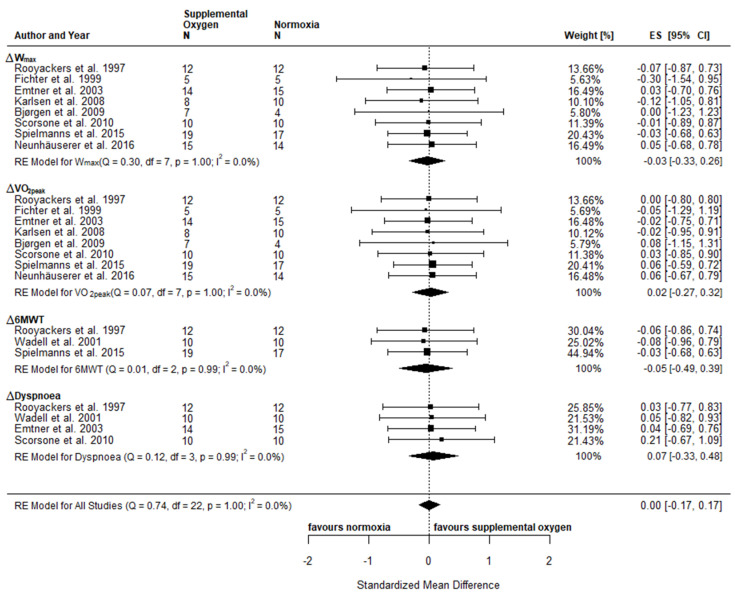
Physical fitness and patient-reported well-being comparing supplemental oxygen and normoxic exercise training interventions using the change from pre to post intervention. CI confidence intervals; ES effects size Cohen’s d (corrected for small samples); df degrees of freedom; I^2^ and Q (Cochran’s Q) describe heterogeneity, RE random effects model.

**Figure 6 jpm-10-00135-f006:**
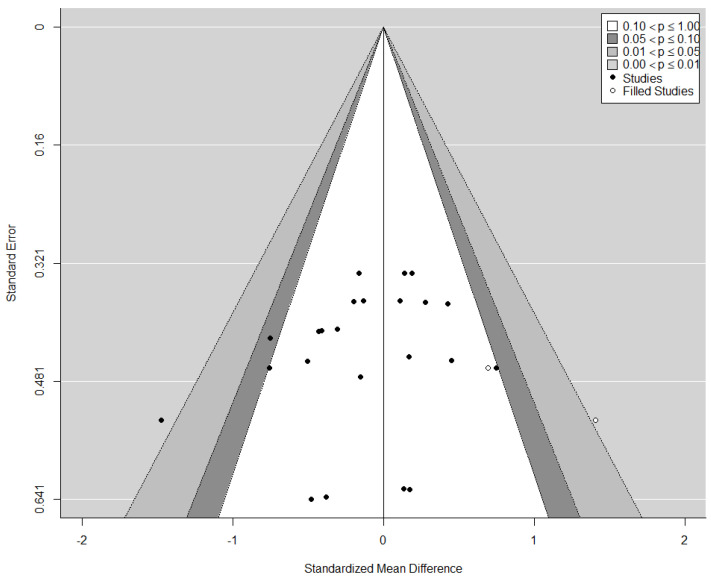
Funnel plot for publication bias assessment including the trim-and-fill-function to plot potentially missing publications as well as the contour-function to visualize a significance threshold.

**Table 1 jpm-10-00135-t001:** Characteristics of included studies (chronological order of publication date). Values presented as mean ± SD or median (range) as indicated.

Study	Entity	Design & Blinding	Final Sample & Dropout Rate	Technical Realization	Training Duration & Frequency	Intervention Characteristics	Outcomes of Interest	Conclusion
Rooyackers et al. 1997 Netherlands	COPD	RCT (parallel) Blinding: not reported	SO *n* = 2 women and 10 men; Age: 63 ± 5 yrs. Normoxia: *n* = 2 women and 10 men; Age: 59 ± 13 yrs. Dropouts: not reported	SO: 4 L min^−1^ via dual prong nasal cannula Normoxia: ambient air	10 weeks 5 x week	Supervised interval training: 5 times of 2 min active cycling and 2 min rest; rowing for 5 min; dynamic resistance exercise for arms, shoulders, back and legs for 10 min; isometric resistance exercise for 3 min; functional tests/training i.e., stair climbing, chair rise, slalom walk, carrying light weight (1–2 kg) above and below shoulder level for 13 min (workload was increased as tolerated and not to fall below an SaO_2_ of 90%; warm-up and cool down not specified). Total training time: 51 min	W_max_ VO_2peak_ HR_peak_ 6MWT TTE Dyspnoea Stair climb Weight lifted CRDQ	Supplemental oxygen did not show an advantage compared to normoxia for chronic improvements in QoL, 6MWT, stair-climbing and weight-carrying test. The normoxia condition but not supplemental oxygen led to a statistically significant increase in W_max_ without a group difference.
Fichter et al. 1999 Germany	COPD	RCT (parallel) Blinding: participants	SO: *n* = 5 men; Age: 58 ± 11 yrs. Normoxia: *n* = 5 men; Age: 59 ± 7 yrs. Dropouts: not reported	SO: humidified 35% O_2_ via mixed chamber Normoxia: humidified decompressed air 21% O_2_	4 weeks 5 x week	Supervised constant cycling with 80% of W_max_ for 45 min (warm-up and cool down not specified). Total training time: 45 min	W_max_ VO_2peak_ HR_peak_ BLa	Only the normoxia condition led to a statistically significant higher maximum power output, accompanied by statistically significant lower blood lactate concentrations after the intervention. No statistically significant changes occurred in the supplemental oxygen group.
Garrod et al. 2000 England	COPD	RCT (parallel) Blinding: Participants	SO: *n* = 13; Age: 64.3 (54–77) yrs. Normoxia: *n* = 12; Age: 71.6 (52–81) yrs. Gender ratio not reported Dropouts: SO = 2 Normoxia = 1	SO: 4 L min^−1^ via nasal cannula and conserving device Normoxia: 4 L∙min^−1^ compressed air	6 weeks 3 x week	Supervised dynamic resistance exercise with external loads for arms and without external loading for lower limbs; fast walking over 10 m at 80% of VO_2peak_ determined by baseline shuttle walk test; unloaded cycling until intolerance (warm-up and cool down not specified). Total training time: 60 min	ISWT Dyspnoea CRDQ HADS LCADL	Supplemental oxygen led to a statistically significant greater reduction in dyspnoea compared to the normoxia condition. Covered distance of the shuttle walk test and patient-reported outcomes (i.e., hospital anxiety and depression scale, chronic respiratory disease questionnaire and London chest activity of daily living scale) improved during a rehabilitation program of 6 weeks without any statistical differences between the two conditions.
Wadell et al. 2001 Sweden	COPD	RCT (parallel) Blinding: participants	SO: *n* = 6 women and 5 men; Age: 65 (52–73) yrs. Normoxia: *n* = 6 women and 5 men; Age: 69 (60–72) yrs. Dropouts: SO = 1 Normoxia = 1	SO: 5 L min^−1^ via dual prong nasal cannula Normoxia: 5 L min^−1^ via dual prong nasal cannula of compressed air	8 weeks 3 x week	Supervised interval training on a motorized treadmill: 5 min warm-up, 2–3 min higher speed separated by 2–3 min lower speed, 2–5 min cool-down; intensity was set to achieve target dyspnoea 7 out of 10 or subjective exertion of 17 on the 6–20 BORG scale; training was paused if SaO_2_ fall below 90% or values of dyspnoea and exhaustion exceeded target range. Total training time: 30 min	HR_peak_ BLa 6MWT RPE Dyspnoea	Distance covered in the 6MWT improved statistically significant in both groups with a greater increase in the normoxia condition. Furthermore, only the normoxia group statistically decreased rating for perceived exertion in the 6MWT.
Emtner et al. 2003 United States of America	COPD	RCT (parallel) Blinding: Participant, exercise supervisors and analyzing investigators	SO: *n* = 6 women and 8 men; Age: 66 ± 7 yrs. Normoxia: *n* = 5 women and 10 men; Age: 67 ± 10 yrs. Dropouts: SO = 1	SO: 3 L min^−1^ via nasal cannula Normoxia: 3 L min^−1^ via nasal cannula of compressed air	7 weeks 3 x week	Supervised ergometer cycling: 5 min warm-up, 35 min of high-intensity, 5 min cool-down. Exercise intensity during the first week was low and increased to 75% of W_max_ (determined by baseline incremental test) in second week. Further adjustments were made according to dyspnoea and fatigue sensation. Total training time: 45 min	W_max_ VO_2peak_ HR_peak_ BLa TTE Dyspnoea Leg fatigue CRDQ SF-36	Both groups statistically improved exercise tolerance with a higher increase in total work rate in the supplemental oxygen group compared to the normoxia condition. Supplemental oxygen caused a greater tolerance to high-intensity exercise compared to high-intensity loading in ambient condition.
Karlsen et al. 2008 Norway	CAD	RCT (parallel) Blinding: not reported	SO: *n* = 2 women and 6 men; Age: 61.1 ± 7.1 yrs. Normoxia: *n* = 3 women and 7 men; Age: 63.6 ± 6.5 yrs. Dropouts: SO = 2 Normoxia = 1	SO: 100% O_2_ enriched air via Douglas bag connected to a tank via a face mask and three valve system Normoxia: ambient air	10 weeks 3 x week	Supervised treadmill walking: 5 min warm-up, high-intensity training of 4 × 4 min at 85–95% HR_peak_ separated by 3 min active rest at 60–70% HR_peak._ Treadmill speed and inclination where increased throughout the study period (cool-down not reported). Total training time: 31 min	W_max_ VO_2peak_ HR_peak_ BLa RPE SF-36 Macnew	Exercise performance, VO_2peak_, maximal ventilation and cardiac output as well as stroke volume increased statistically significant in both groups without a between-group effect. Patient-reported outcomes improved similarly. Supplemental oxygen did not show any additional training effects.
Bjørgen et al. 2009 Norway	COPD	RCT (parallel) Blinding: not reported	SO: *n* = 5 women and 2 men; Age: 61 ± 12 yrs. Normoxia: *n* = 2 women and 3 men; Age: 61 ± 8 yrs. Dropouts: SO = 1 Normoxia = 2	SO: 100% O_2_ via mask connected to a plastic balloon which was constantly refilled Normoxia: ambient air	8 weeks 3 x week	Supervised one-legged interval cycling: 10 min warm-up (both legs); high-intensity training of 8 × 4 min at 85–95% HR_peak_ (legs alternating); the resting leg was placed on the bike between the pedals (cool-down not reported). Total training time: 42 min	W_max_ VO_2peak_ HR_peak_ BLa RPE	Exercise performance improved statistically significant in both conditions without a between-group difference. Supplemental oxygen did not show any additional training effects compared to ambient air.
Scorsone et al. 2010 Italy	COPD	RCT (parallel) Blinding: participants and exercise supervisors	SO: n = 3 women and 7 men; Age: 67 ± 9 yrs. Normoxia: *n* = 3 women and 7 men; Age: 68 ± 7 yrs. Dropouts:none	SO: 40% O_2_ through mouthpiece connected to a Douglas bag Normoxia: humidified room air through mouthpiece connected to a Douglas bag	8 weeks 3 x week	Supervised ergometer cycling: 5 min warm-up, 30 min high-intensity exercise, 5 min cool-down. Exercise intensity during the first week was 40% of W_max_ and increased to 80% of W_max_ (determined by baseline incremental test) in second/third week. Adjustments were made according to dyspnoea and fatigue sensation. Total training time: 40 min	W_max_ VO_2peak_ HR_peak_ TTE Dyspnoea Leg fatigue	High-intensity training led to statistically significant improvements of VO_2peak_, W_max_ and time to exhaustion without a between-group difference. Supplemental oxygen did not show any additional training effects compared to humidified normoxic air.
Ringbaek et al. 2013 Denmark	COPD	RCT (parallel) Blinding: no	SO: *n* = 11 women and 11 men; Age: 69.4 ± 9.8 yrs. Normoxia: *n* = 10 women and 13 men; Age: 68.6 ± 7.8 yrs. Dropouts: Week 0–7 SO = 6 Normoxia = 1 Week 7–20 SO = 1 Normoxia = 1	SO: 2 L min^−1^ through a 2.3 kg portable oxygen concentrator including a conserving deviceNormoxia: ambient air	7 weeks 2 x week Followed by 13 weeks 1 x week	Supervised walking and cycling: 30 min including warm-up and cool-down. Exercise intensity was set to reach 85% of VO_2peak_ (determined by baseline incremental shuttle walk test). Total training time: 30 min	ESWT SGRQ	Both groups improved statistically significant endurance performance assessed via endurance shuttle walk test at 85% VO_2peak_ (predicted through incremental shuttle walk test) and patient-reported outcomes (St. George’s Respiratory Questionnaire) without a between-group effect. Supplemental oxygen showed no additional benefits compared to ambient air.
Spielmanns et al. 2015 Germany	COPD	RCT (parallel) Blinding: participants and investigator	SO: *n* = 9 women and 10 men; Age: 65 ± 8.7 yrs. Normoxia: *n* = 8 women and 9 men; Age: 64 ± 8.4 yrs. Dropouts: SO = 23 Normoxia = 26	SO: 4 L min^−1^ O_2_ via nasal cannula leading to an FiO_2_ of approximately 0.35 Normoxia: compressed air at 4 L min^−1^ via nasal cannula	24 weeks 3 x week	Supervised ergometer cycling: First 12 weeks included an interval training and the second 12-week period included a continuous loading. Exercise intensity for interval training: weeks 1–3 with 6 × 1 min at 110% W_max_ and 4 min at 60% W_max_ as active rest; weeks 4–6 with 6 × 1 min at 115% W_max_ and 4 min at 65% W_max_; weeks 7–9 with 6 × 1 min at 120% W_max_ and 4 min at 70% W_max_; weeks 10–12 with 6 × 1 min at 125% W_max_ and 4 min at 75% W_max_. Continuous loading increased every 3 weeks by 5% starting from 70% W_max_ to 85% W_max_. Total training time: 30 min	W_max_ VO_2peak_ BLa 6MWT SF-36	Both groups showed statistically significant improvements in QoL, exercise tolerance, VO_2peak_ and distance covered in the 6MWT. Apart from further increase in exercise tolerance, these improvements occurred within the first 12 weeks and were maintained in week 24. Supplemental oxygen did not show any further enhancing effects on outcome values.
Neunhäuserer et al. 2016 Austria	COPD	RCT (parallel and cross over ^†^) Blinding: participants and investigator	SO: *n* = 1 women and 14 men; Age: 63.1 ± 5.4 yrs. Normoxia: *n* = 7 women and 7 men; Age: 64.1 ± 6.1 yrs. Dropouts before crossover (total): SO = 6 (7) Normoxia = 7 (8)	SO: 10 L min^−1^ O_2_ via nasal cannula leading to an FiO_2_ of approximately 0.6 Normoxia: compressed air at 10 L min^−1^ via nasal cannula	6 weeks ^‡^ 3 x week	Supervised ergometer interval cycling and resistance training: 5-min warm-up; 7 × 1 min at 70–80% W_max_ and 2 min at 50% W_max_ as active rest; 5 min cool-down. Resistance training included 8 high-intensity exercises performed on machines with 1 set of 8–15 reps to failure: latissimus pull-down, shoulder press, back extension, abdominal crunch, butterfly, reverse butterfly, leg extension and leg flexion. Progression was made if more than 15 reps were performed. Total training time: 31 min without time of resistance training	W_max_ VO_2peak_ HR_peak_ BLa 10-RM ^§^ HADS ^§^	Supplemental oxygen showed statistically significant improvements in relative W_max_ compared to normoxia. Strength gains increased in both groups without a significant between-group effect. Supplemental oxygen added superior effects on top of the endurance-induced improvements without an effect on muscle strength.
Alison et al. 2019 Australia	COPD	RCT (parallel) Blinding: participants, therapists and investigators	SO: *n* = 28 women and 30 men; Age: 69 ± 7 yrs. Normoxia: *n* = 22 women and 31 men; Age: 69 ± 8 yrs. Dropouts: SO = 6 Normoxia = 8	SO: 5 L min^−1^ via dual prong nasal cannula delivered via an oxygen concentrator producing 90 ± 3 % oxygen Normoxia: 5 L min^−1^ via dual prong nasal cannula of medical air	8 weeks 3 x week	Supervised treadmill walking: 20 min at 80% average speed assessed by 6MWT and 10 min cycling at 60% W_max_ estimated from 6MWT with progression to 20 min cycling by week 3 leading to a total training time of 40 min. Intensity was modified to keep dyspnoea and RPE between 3 and 4 on the modified BORG scale (0–10). Total training time: 30–40 min	ESWT ISWT CRDQ Dyspnoea-12	Supplemental oxygen and normoxia statistically improved statistically exercise capacity assessed by ESWT and ISWT as well as quality of life in COPD patients who demonstrated oxygen desaturation during exercise. No statistical between-group difference were observed for dyspnoea, although dyspnoea statistically improved only in the supplemental oxygen group.

*6MWT* 6-min walking test; *10-RM* ten repetition maximum; *BLa* blood lactate concentration; *CAD* coronary artery disease; *cm* centimetre; *COPD* chronic obstructive pulmonary disease; *CRDQ* Chronic Respiratory Disease Questionnaire; *Dyspnoea-12* Dyspnoea-12 questionnaire; *ESWT* endurance shuttle walk test; *FiO_2_* inspired oxygen fraction; *FU1* first follow-up; *FU2* s follow-up; *h* hours; *HADS* Hospital Anxiety and Depression Scale; *HR_peak_* peak heart rate; *ISWT* incremental shuttle walk test; *kg* kilogram; *L* liters; *LCADL* London Chest Activity of Daily Living Scale; *LoE* level of evidence; *Macnew* Heart Disease Health-related Quality of Life Questionnaire; *O_2_* oxygen; *QoL* quality of life; *RCT* randomized controlled trial; *RPE* rating of perceived exertion; *SaO_2_* oxygen saturation; *SF-36* 36-Item Short Form Health Survey; *SGRQ* St. George’s Respiratory Questionnaire; *SO* supplemental oxygen; *TTE* time to exhaustion; *VO_2peak_* peak oxygen consumption; *W_max_* maximal watt; *yrs.* Years. NOTE: † only the midpoint comparison before crossover was used for evaluation and, thus, the crossover comparison of Neunhäuserer et al. was excluded, ‡ study period was actually 22 weeks including a 6-week training free initial period and 1-week training free period used for assessments between the crossover, § no data provided before crossover.

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
