# Peer review of "Is Structured Exercise Performed with Supplemental Oxygen a Promising Method of Personalized Medicine in the Therapy of Chronic Diseases?"

_jpm, 2020, doi:10.3390/jpm10030135_

Round 1

Reviewer 1 Report

This is a very interesting systematic review with great clinical significance. It's well written and methodologically sound. My only concern is the fact that the hyperbaric oxygen treatments have been eliminated from the analysis. HBO therapy is a very promising area of research in both athletes and patients. I would like to see this component in the current review, however, this is not feasible at the moment due to the exclusion criteria applied by the authors. 

I don't have any particular major comments to make. I really enjoyed reading this manuscript. 

Author Response

Thank you for your very positive feedback on our paper. Please let us clarify the point you made concerning the exclusion of HBO. Actually, we did not exclude studies that investigate the effects of combined hyperbaric oxygen and exercise therapy, but we did not identify any papers on this in our systematic literature search. Lines 148 to 151 states that excluded articles of HBO were reviewed for possible further literature references. This could have been somewhat misleading, since this refers only to studies using sole HBO without an exercise component. We have now amended this part to improve readability.

Reviewer 2 Report

The article 'Is structured exercise with supplemental oxygen a promising method
 of personalized medicine in the therapy of chronic diseases? ' is a very good attempt to summarize and evaluate the existing research. The study is  comprehensive, however, the authors should address following minor concerns 

1- The authors should elaborate the pros and cons of supplemental oxygen therapy further in introduction section. 

2- The authors have discussed hypoxia and hypoxemia. There are several key research papers related to the level of oxygen in blood which may be cited to further explain the concept. For example the research group of John Kheir ,  https://doi.org/10.1002/adhm.201200350 and some other researchers including Khan et al https://doi.org/10.1080/21691401.2018.1492420
have discussed the  concepts of hypoxia and hypoxemia. 

3- The authors should mention how was the oxygen saturation of patients in various studies monitored? It is important because the level of hypoxia /hypoxemia and supplemental oxygen are related. 

Author Response

The article 'Is structured exercise with supplemental oxygen a promising method of personalized medicine in the therapy of chronic diseases? ' is a very good attempt to summarize and evaluate the existing research. The study is comprehensive, however, the authors should address following minor concerns:

1 - The authors should elaborate the pros and cons of supplemental oxygen therapy further in introduction section.

Thank you for this valuable advice. Further information have been added to the corresponding section in the introduction.

2 - The authors have discussed hypoxia and hypoxemia. There are several key research papers related to the level of oxygen in blood which may be cited to further explain the concept. For example the research group of John Kheir https://doi.org/10.1002/adhm.201200350 and some other researchers including Khan et al https://doi.org/10.1080/21691401.2018.1492420 have discussed the concepts of hypoxia and hypoxemia.

Thank you for pointing this out. As hypoxia and hypoxemia are hallmarks of chronic diseases and inflammation as well as key factors for decreased exercise performance, we attempted to discuss these issues within the paper. However, the scope of this review was not to comprehensively discuss the underlying mechanisms of tissues hypoxia but rather to evaluate the efficacy of supplemental oxygen to enhance performance while simultaneously counteracting hypoxia (although it has not been measured in the studies identified). Your suggested references are quite interesting, although there are currently no trials examining the efficacy of oxygen nanobubbles within an exercise therapy approach. Nevertheless, we added this valuable information within the discussion section as we share your opinion that it bears a huge potential for a personalized medicine setting. Thank you.

3 - The authors should mention how was the oxygen saturation of patients in various studies monitored? It is important because the level of hypoxia /hypoxemia and supplemental oxygen are related.

Thank you for this important comment. In fact, we were also interested in the oxygen saturation levels but noticed a large variation in monitoring and/or reporting within the included studies. In some studies oxygen saturation was not assessed due to blinding, while in others it was not mentioned at all or measured only at exercise testing. Therefore, we decided not to include this data since it was also not part of our main outcomes. 
